# Advances in Purification, Modification, and Application of Extracellular Vesicles for Novel Clinical Treatments

**DOI:** 10.3390/membranes12121244

**Published:** 2022-12-08

**Authors:** Yasunari Matsuzaka, Ryu Yashiro

**Affiliations:** 1Division of Molecular and Medical Genetics, Center for Gene and Cell Therapy, The Institute of Medical Science, University of Tokyo, Minato-ku, Tokyo 108-8639, Japan; 2Administrative Section of Radiation Protection, National Institute of Neuroscience, National Center of Neurology and Psychiatry, Kodaira, Tokyo 187-0031, Japan; 3Department of Infectious Diseases, Kyorin University School of Medicine, 6-20-2 Shinkawa, Mitaka-shi, Tokyo 181-0004, Japan

**Keywords:** exosomes, extracellular vesicles, mesenchymal stem cells, miRNA

## Abstract

Extracellular vesicles (EV) are membrane vesicles surrounded by a lipid bilayer membrane and include microvesicles, apoptotic bodies, exosomes, and exomeres. Exosome-encapsulated microRNAs (miRNAs) released from cancer cells are involved in the proliferation and metastasis of tumor cells via angiogenesis. On the other hand, mesenchymal stem cell (MSC) therapy, which is being employed in regenerative medicine owing to the ability of MSCs to differentiate into various cells, is due to humoral factors, including messenger RNA (mRNA), miRNAs, proteins, and lipids, which are encapsulated in exosomes derived from transplanted cells. New treatments that advocate cell-free therapy using MSC-derived exosomes will significantly improve clinical practice. Therefore, using highly purified exosomes that perform their original functions is desirable. In this review, we summarized advances in the purification, modification, and application of EVs as novel strategies to treat some diseases.

## 1. Introduction

Extracellular vesicles (EVs) were discovered in 1946 [1]. Secretory vesicles with a diameter of approximately 100 nm, discovered in 1981 during reticulocyte research, were first named exosomes in 1987 [2]. Subsequently, for a long time, exosomes were thought to be a part of intracellular waste disposal mechanism [3]. However, exosomes contain messenger RNA (mRNA) and microRNA (miRNA) derived from secretory cells, and information exchange takes place between cells when exosomes are transported to other cells [4,5,6]. Since the amount and type of functional molecules, such as RNAs and proteins, in exosomes or membranes vary with diseases, they are suitable for disease diagnosis, prognosis, and identification of therapeutic targets [7,8]. However, since exosome composition does not always match that of secretory-derived cells, the mechanism by which specific molecules are sorted into luminal vesicles in multivesicular bodies remains largely unknown. However, several RNA-binding proteins were found in exosomes [9,10]. Since exosomes are regarded as natural drug delivery systems, they are widely used as a drug discovery technology [11,12]. Furthermore, since exosomes exist in many species, the possibility of information transmission across species has also been suggested, and studies are being conducted to elucidate the mechanisms of various life phenomena and to broadly apply them in the health and medical fields [12,13,14,15,16].

Extracellular vesicle is a general term for membrane vesicles surrounded by a lipid bilayer that is secreted by various tissues and cells [17,18,19,20,21,22,23,24,25,26,27]. These vesicles are classified into exosomes (40–200 nm) that are secreted outside the cell, microvesicles (200–1000 nm) that directly bud from the cell membrane and are secreted outside the cell, and apoptotic vesicles (1000–5000 nm) that are secreted from apoptotic cells [24,28,29,30,31,32,33,34,35]. However, identifying the origin of extracellularly secreted EVs is challenging. Since the classification of EVs is vague, the International Society for EVs (ISEV) has taken the lead to organize EVs [36,37,38,39]. To separate EVs in a pure form from other EVs, such as microvesicles, using centrifugal force is challenging. Categorization into small (ultracentrifugation 100,000× *g* pellet fraction, particles of size equivalent to exosomes sediment), medium (medium speed centrifugation 20,000× *g* pellet), and large EVs (low-speed centrifugation 2000× *g* pellet) has been proposed [40]. Furthermore, through analysis of EVs fractionated according to their size, they were classified into exomeres, without a lipid bilayer, small exosomes, and large exosomes [41,42,43,44,45,46,47,48]. Since exosomes refer to the sum of exosomes released by various cells, only their average values can be analyzed. For example, when exosomes are isolated from peripheral blood to identify those that are diseased and cell derived, the biomarker detection sensitivity is low because healthy cell-derived exosomes are present in majority. Therefore, establishing a method for sorting exosomes using markers specific to organs and cells is desirable. Furthermore, the technology of analyzing exosomes at the level of one particle per cell is still insufficient; however, this technology is hypothesized to lead to the development of highly sensitive diagnostic methods [49,50,51]. In addition, clearly distinguishing exosomes from other EVs is challenging, and high sensitivity can be achieved only if exosomes are highly pure at the time of analysis.

## 2. Constitution and Characterization of Exosomes

Exosomes are EVs produced from multivesicular endosomes during cell endocytosis and are extracellularly secreted [52,53,54,55,56,57,58,59,60,61,62]. Exosomes are internally formed by budding within endocytic compartments through the fusion of plasma membranes and vesicle-containing endosomes [63,64,65,66]. They are isolated from body fluids such as blood, urine, saliva, and cerebrospinal fluid and contain various biomolecules such as proteins, nucleic acids, and lipids, which are important for intercellular communication [67,68,69,70,71,72,73,74,75,76,77,78,79,80]. Therefore, EVs have attracted attention as biomaterials with drug-delivery capabilities. Owing to their endogenous origin, exosomes are less likely to be immunogenic or cytotoxic compared with that of other artificial delivery agents [59,81,82,83,84,85,86,87,88,89,90,91,92,93,94,95,96,97,98,99,100,101,102,103,104,105,106,107,108,109]. In addition, the lipid bilayers of exosomes can protect drugs from rapid blood clearance and reduce unintended drug-induced cytotoxicity. Notably, the exosome is large enough to prevent rapid renal clearance and small enough to avoid being absorbed through the reticuloendothelial system [110]. Small nanoparticles, such as exosomes, tend to accumulate in cancerous tumor sites owing to leaky blood vessels and abnormal lymphatic drainage; thus, exosomes are ideal for drug delivery to treat certain cancers [78,87,93,111,112,113,114,115,116,117,118,119,120,121,122,123,124,125,126,127,128,129]. Similar to other vesicles, exosomes consist of a lipid bilayer membrane with an aqueous inner compartment and a lipophilic outer layer. This structure allows both hydrophobic and hydrophilic drugs to be loaded into exosomes. Exosomes are of different types; although they contain different amounts of cellular components, certain lipids, proteins, and nucleic acids are common components [8,21,71,72,75,79,85,91,92,102,112,130,131,132,133,134,135,136,137,138,139,140,141,142,143,144,145,146,147,148,149,150,151,152,153,154,155,156,157,158,159,160,161,162,163,164,165,166,167,168,169,170,171,172,173,174,175,176,177,178,179,180]. Exosomes contain large amounts of cholesterol, sphingolipids, phosphoglycerides, ceramides, and saturated fatty acids, which bind with each other to contribute to exosome stability, and a variety of membrane-bound and intracellular proteins [181,182,183,184,185,186,187,188,189,190]. The most common molecules are membrane transporters and fusion proteins, major histocompatibility complexes, heat shock proteins, tetraspanins, endosomal transport sorting complexes required for transport (ESCRTs), and lipid raft-associated proteins [109,134,191,192]. In addition, exosomes are rich in proteins that are specific to the cell types that contain them. For example, exosomes extracted from dendritic cells are rich in a heat shock protein (Hsp73) that may independently exert the anti-tumor effects observed in the whole exosomes [193]. In addition, exosomes contain nucleic acids such as miRNAs, non-coding RNAs, and mRNAs [194,195,196,197,198,199]. Notably, these RNAs are found within the aqueous compartment of exosomes and are bound to the outer membrane of exosomes via the protein Argonaute 2 (Ago2) [200]. Specific miRNAs in exosomes can be targeted using proteins that recognize short RNA motifs, such that RNAs can be selectively packaged into exosomes (Figure 1) [201,202,203,204,205,206,207]. 

Further, exosomes are released from various cell types, providing a wide range of donor cell options to isolate exosomes. Two important factors in determining donor cells are the biological characteristics of exosomes and amount of exosomes extracted from a particular cell type. For example, exosomes of dendritic cells stimulate stronger anti-tumor immune responses compared with those of exosomes from EG7 tumor cells; exosomes of dendritic cells promote proliferation and differentiation of T cells or may contain more molecular factors that more efficiently interact with T cells compared with those of exosomes extracted from tumor cells [208]. Mesenchymal stem cells (MSCs) produce many exosomes, which are easy to isolate, can be cultured in large quantities, and promotes cell viability [209,210]. They have been identified as a particularly promising cell type. Whether cell culture media or body fluids are used, the purity and quantity of exosomes are extremely important for the development of exosome-based drug delivery agents. Exosomes are typically purified using differential ultracentrifugation and quantified using protein assays [211,212,213,214]. Other methods include filtration, immunoaffinity isolation, and microfluidic analytical techniques to rapidly isolate exosomes for structural and physical analyses; however, whether exosomes purified using these new methods are effective for drug delivery is unclear [215,216,217,218]. Using in vivo or ex vivo techniques, therapeutic agents can be loaded into exosomes. Various ex vivo techniques, such as freeze–thaw cycles, saponin membrane permeabilization, sonication, and extrusion, have been used to load drugs into exosomes [218,219,220,221]; in these techniques, drug behavior and activity were maintained after loading the drugs into exosomes. Additionally, exosomes are stable when stored at −20 °C to −80 °C and exposed to several freeze–thaw cycles [222,223]. 

## 3. Molecular Regulation and Biomarkers with Exosome in Diseases

The most advanced research on the functions and applications of exosomes has been in the field of cancer. For example, pancreatic cancer-derived exosomes were shown to make normal cells malignant, and the exosomal proteins, secreted by pancreatic cancer cells, that were responsible for inducing malignancy were identified [178,224,225,226]. Exosome-encapsulated miRNAs released from cancer cells induce angiogenesis within tumors and are involved in cancer proliferation and metastasis [227,228,229,230]. Exosomes secreted from metastatic cancer cells are enriched in specific proteins that promote the formation of a premetastatic niche [178,231,232,233,234]. In addition, miR-155, which is encapsulated in exosomes secreted by breast cancer cells, promotes the formation of beige and brown adipocytes, induces metabolic remodeling by suppressing Pparg expression, and is involved in cancer cachexia [235]. Exosomes are involved in carcinogenesis and malignant transformation [236,237]. In neurodegenerative diseases, abnormally aggregated proteins, which are causative proteins, are released outside the cells by exosomes and spread to surrounding cells [238]. Further, exosomes also deliver the viral genome and proteins to the cells surrounding the infected cells during the process of virus propagation between cells in vivo, which is advantageous for viral survival [239]. In the immune system, they are involved in various immune function controls, such as the exchange of antigen information between immune cells and activation/inactivation of immune cells [240,241]. Exosomes are involved in various diseases, and since the types and amounts of functional molecules encapsulated in exosomes or present in the membrane vary depending on the disease, they are highly likely to be applied in disease detection and prognosis and used as therapeutic targets [242,243,244]. Functional molecules, such as RNAs and proteins, that are encapsulated in exosomes or expressed on membranes are stably retained, and their amounts and types vary depending on the disease. For example, in patients with myalgic encephalomyelitis and chronic fatigue syndrome, which are challenging to diagnose, the amount of EVs in plasma increases, and actin network proteins such as talin-1 and filamin A, which are characteristically encapsulated, may serve as biomarkers [245,246,247]. In addition, since neuron-derived exosomes contain substances such as tau, α-synuclein, and TDP-43 that aggregate within neurons, they contain proteins that cause the onset of Alzheimer’s disease, Parkinson’s disease, and amyotrophic lateral sclerosis and are drawing attention for their association with the onset of each disease [238,248,249,250,251,252,253]. Currently, cerebrospinal fluid collected by lumbar puncture is used for research and diagnosis using brain-derived exosomes [254]. In particular, exosome-encapsulated proteins derived from disease-related cells leaked into body fluids, such as α -synuclein in Parkinson’s disease, function as biomarkers of a disease and its severity [132,238,248,249]. When marker candidates useful for early diagnosis and stratification of diseases are identified, verification based on immunoassay methods such as ELISA is performed. However, analysis of endogenous proteins has not sufficiently progressed compared with that of exosome membrane proteins. This is because of challenges associated with immunoassays in which antibodies react with encapsulated proteins and antigens are enclosed within the membrane of exosomes.

In addition, since some brain-derived exosomes are also detected in peripheral blood, neural cell adhesion molecule (NCAM-1) and L1 cell adhesion molecule (L1CAM) were reported as markers [255,256]. Furthermore, exosomes in urine are new diagnostic markers for kidney, prostate, and bladder diseases, those in cerebrospinal fluid are markers for tumors and neurodegeneration in the brain, and those in amniotic fluid are markers that reflect fetal status [69,257,258,259,260,261]. In addition, since exosomes function as disease mediators, suppression of exosome secretion is attracting attention as a new therapeutic method. For example, in drug repositioning of existing drugs, to control the production and secretion of exosomes, a high-throughput screen of 4580 compounds was performed in prostate cancer cells, and five exosome production inhibitor and six exosome production activator candidate compounds were identified [262]. Furthermore, a study that aimed at cancer-specific regulation identified that miR-26a and its regulatory genes *SHC4*, *PFDN4*, and *CHORDC1* regulate exosome secretion by prostate cancer cells [263]. Thus, techniques that can specifically manipulate and modify functional molecules encapsulated in exosomes will contribute to exosome function elucidation and effective functional expression. 

In addition, the effect of mesenchymal stem cell (MSC) therapy, which is being applied in regenerative medicine owing to its ability to differentiate into various cells, is due to humoral factors, including mRNA, miRNAs, proteins, and lipids, which are encapsulated in exosomes derived from transplanted cells [264,265,266]. Exosomes derived from MSCs suppress tissue fibrosis in liver and kidney diseases and are also effective in treating heart and Alzheimer’s diseases [89,267,268]. In addition, clinical trials were conducted to verify the neuroprotective and anti-fibrosis effects of exosomes secreted by umbilical cord-derived MSCs after cochlear implant surgery and to evaluate the wound healing effect of platelet-derived exosomes [269].

## 4. Applications of Exosome as Novel Drug Delivery System 

With the recognition of the importance of EVs, development of techniques that individually isolate them and analyzing the molecules contained within hold high significance. The transfer of the contents of exosomes to other cells has been analyzed using DNA, RNA, proteins, and intercellular signaling substances. Although the detailed delivery mechanism is of high significance, it has not yet been elucidated. A study reported that EVs specifically adhere to cells, and the development of observational techniques, including live-cell imaging, is necessary to clarify this phenomenon. Clarifying the delivery mechanism of the cargo of EVs will not only lead to true understanding of the types of organisms affected by EVs but also provide clues to the control of substance transport by EVs; the results will also be useful in designing artificial particles that mimic EVs. A device that isolates exosomes in a fluid using microfabrication technology and a method that attaches a tag, such as a barcode, to isolate them are being tested [270,271]. The method of analyzing molecules contained in a single vesicle is challenging, and the extension of conventional omics technology has limitations. An enhanced Raman scattering method using a metal near field and a device that analyzes the surface proteins of individual extracellular microparticles using fluorescent antibodies is under development [272]. In addition, live imaging is a powerful method for understanding the mechanism of EV formation. Super-resolution imaging is essential for observing exosomes with a diameter of ≤200 nm. Its application to living cells was a challenge; however, high-speed super-resolution live imaging technology has been developed [273]. In addition, a technique was developed for the real-time observation of extracellularly released microparticles using a single-cell observation chip [274]. The development of imaging technology is indispensable for the dynamic analysis of structures of several nanometers. This requires the development of microscopes and techniques associated with observing samples, such as sample-fixation techniques and microfluidic devices.

Unlike artificial products such as liposomes, exosomes can function as a natural drug delivery system (DDS), delivering siRNA, miRNA, and low-molecular-weight compounds to target cells. The in vivo dynamics of exosomes is not yet elucidated, and elucidation of the molecular mechanism that controls the dynamics has the potential to contribute to the development of DDSs with high target accuracy. Various cell adhesion molecules and sugar chains are expressed on the surface of exosomes, and based on the expression pattern, the cells whose exosomes exhibit affinity are identified [275,276]. Further, new DDSs are being developed by modifying and applying the properties of exosomes. For example, by encapsulating a STING agonist in exosomes, in which a therapeutic protein such as prostaglandin F2 receptor negative regulator (PTGFRN), is highly expressed in the exosome membrane, ExoSTING was developed to activate the STING pathway in cancer in an antigen-presenting cell-specific manner [277]. Furthermore, clinical trials of exoIL-12, in which IL-12 locally acts on cancer cells through exosomes (with IL-12 expressed on their surface), were conducted (Figure 2) [278]. In addition, bovine milk-derived exosomes are being developed as carriers for oral administration of nucleic acids, peptides, and small molecules, which are challenging to be orally administered [279].

In addition, delivering drugs across certain biological barriers, especially the blood–brain barrier (BBB), is a big challenge in chemotherapy. For example, exosomes released by glioblastomas have been detected in the serum, suggesting that endogenous exosomes cross the BBB [280]. Furthermore, in experiments with mice, exosome preparations loaded with the anti-inflammatory polyphenol curcumin were delivered through the nasal cavity, inducing apoptosis of follicle cells, suggesting that vesicle preparations cross the BBB [281]. In addition, exosomes extracted from brain epithelial cells and loaded with anticancer agents cross the BBB and induce cytotoxicity in zebrafish tumor cells [282]. Taken together, these results suggest that exosomes are particularly effective for drug delivery to the brain. 

## 5. Purification Methods and Drawbacks of Exosome

Although EV-based drug delivery appears to be a promising and effective treatment, several major challenges need to be resolved before they can be safely and efficiently introduced for clinical applications. First, EV isolation and purification procedures should be standardized to eliminate contaminants such as protein aggregates and to improve reproducibility. Second, identifying donor cells that are stable sources of EVs and fully characterizing the EVs extracted from these cells are essential. Finally, developing highly efficient methods for loading drugs into EVs to maximize drug delivery efficacy is necessary. While studies on the identification of biomarkers for diseased cell-derived exosomes and development of diagnostic methods using them are being actively conducted, little progress has been made in the development of methods to particularly isolate diseased cell-derived exosomes. By identifying surface markers specific to exosomes derived from diseased cells, the development of a removal method that does not affect exosomes derived from normal cells is desired. To use exosomes as therapeutic agents and DDS tools, large amounts of high-quality exosomes have to be prepared. As the condition of the producing cells also affects the quality of exosomes, strict culture conditions have to be set. In addition, application of techniques that increase the amount of exosomes produced is recommended. Furthermore, implementing isolation and purification methods using ultracentrifugation on an industrial scale is challenging, and the establishment of other methodologies that can process large amounts of EVs is desired. In addition, establishing evaluation methods and concepts that strictly define the quality of exosomes, such as quantity, purity, particle size, distribution, homogeneity, and potency are necessary.

Thus, various studies are being conducted on EVs, particularly on exosomes; however, a major bottleneck is the lack of standard and efficient techniques for the production and isolation of EVs. When dealing with body fluids, EVs coexist with many proteins and cells with physical and chemical properties similar to that of EVs, making their isolation inherently complex [283,284]. The main separation methods used are those that use differences in EV density, size, and specific surface markers. Techniques based on these principles include ultracentrifugation, precipitation, filtration, size-exclusion chromatography, immunoaffinity, and antigen-antibody reactions [210,211,212,213,214,284,285,286,287,288]. Ultracentrifugation is a separation method that exploits differences in density and size between cells, EVs, and proteins. Separation using an ultracentrifuge is usually used to isolate EVs; however, the drawbacks of this technique are the time taken for collection and high throughput. Cells, apoptotic bodies, and large vesicular fractions of EVs can be separated using standard centrifugation at <20,000× *g*, whereas centrifugation at <100,000× *g* should be used to purify exosomes from proteins (Figure 3) [289,290,291]. The major drawback of this method is that it requires a high spin speed and a long operating time of approximately 5 h. In addition, the ultracentrifugation method has a drawback in that the recovery rate of exosomes is low and protein aggregates co-precipitate. Sucrose density gradient centrifugation was used as an additional technique for ultracentrifugation to improve the isolation purity efficiency of exosomes [292]. The sedimentation method was developed without the ultracentrifugation step and has challenges with recovery time and high throughput. The separation efficiencies of these sedimentation methods were compared with those of conventional ultracentrifugation methods, which indicated high separation efficiencies [293,294]. However, factors such as residual precipitating matrices and polymeric additives can affect the biological activity and properties of EVs, including exosomes. In filtration method, membrane filters (pore size of approximately 50 to 450 nm) are used to separate large vesicle components of cells, including EVs, in biological samples [295]. The filtration method uses a membrane to sieve the large vesicle fractions and EVs, followed by the flow of the small vesicle fractions of EVs and exosomes from the proteins using ultracentrifugation. To separate the small vesicle fraction of EVs/exosomes from protein. aggregates and avoid ultracentrifugation, for molecules of 100 kDa, ultrafiltration is generally used. Filtration is generally faster than ultracentrifugation, but exosome yield may decrease due to clogging effects caused by the non-optimization of the operating procedure [296,297]. In addition, size exclusion chromatography (SEC) was used to separate EVs and exosomes from protein aggregates [298,299,300]. Typically, cells and large vesicle fractions of EVs are removed using centrifugation or filtration, followed by small vesicle fractions and exosomes using a size exclusion column. Small substances, such as proteins, are retained on the column for a long time, while large substances, including small vesicle fractions of EVs and exosomes, elute early. Therefore, the separation of small vesicle fractions and exosomes of EVs can be achieved by collecting fractions that elute at specific times. Immunoaffinity separation is a method for separating exosomes from other EVs using specific surface markers [301,302,303,304,305,306]. Further, EVs and exosomes contain cell-of-origin specific markers, and antigen-antibody reactions can prove to be beneficial. A common immunoaffinity-based separation method utilizes antibody-coated magnetic beads to capture EVs and exosomes that contain specific markers, in bodily fluids. Although this method allows the isolation of specific subfractions of EVs or exosomes, it is generally not suitable for isolating EVs or exosomes from large amounts of biological samples. In addition, by using a molecule that binds to phospholipid phosphatidylserine, unlike in conventional methods such as ultracentrifugation, that is specifically expressed on the surface of exosomes, purifying exosomes with 100 times more purity that conventional methods and detecting exosomes with high sensitivity are possible [52]. Moreover, the conventional separation methods require dedicated laboratory equipment or reagents and multi-step work processes, and analyzing EVs and exosomes as routine diagnostic processes in clinical practice poses many challenges. However, microfluidic systems may overcome these shortcomings; this technology enables rapid isolation and analysis of EVs and exosomes from clinical specimens, facilitating diagnosis and treatment [307,308,309,310,311,312]. Further, the microfluidic system not only provides a multipurpose platform for the isolation and analysis of EVs and exosomes, but also contributes to the integration and simplification of multiple processes and risk reduction of cross-contamination. In addition, various methods of separating particles by employing characteristics such as particle size, shape, and electric charge of fine particles have been explored. 

## 6. Application of Exosome in Plant and Food as DDS 

Furthermore, exosome studies have mainly focused on animals, such as mammals, but plants also release exosome-like EVs [313,314,315,316]. Using plant-derived exosomes, the growth and toxicity of fungi that are harmful to plants can be suppressed, enabling plants to defend themselves [317]. In addition, the functions of exosomes present in food ingested by humans are also being investigated, and ingested plant-derived exosomes affect the composition of the intestinal microbiota and physiological functions of the host [318,319]. 

Further, EVs have been shown to activate immunity in various bacteria [320,321,322,323,324]. Vaccines against infectious diseases, including coronavirus disease 2019, and new immunotherapies with cancer-suppressing effects have been developed using new technologies that contribute to the development of modified EVs [325,326,327]. Although the mechanism is unknown, EVs tend to accumulate in cancer cells and are also attracting attention as a tool in DDSs to target specific cells [328]. Further, EVs are beginning to be used as vaccines and antibiotic transporters worldwide; however, since EVs are heterogeneous, the products used for clinical applications must be highly pure and uniform [242,329,330,331,332]. Therefore, modifying and artificially synthesizing EVs are essential. Further, DDSs that use liposomes and other technologies to encapsulate and localize arbitrary proteins, nucleic acids, and other substances are also essential. Thus, studies on the development of a new artificial membrane particle that mimics EVs are essential. 

## 7. Conclusions

Exosomes, which are representative EVs, have a wide range of applications, from the elucidation of disease mechanisms to diagnosis and treatment, mainly in the field of cancer and neurological diseases. However, exosomes secreted from MSCs have therapeutic effects in various diseases, and studies are conducted to develop new therapeutic agents for these diseases. As an increasing number of studies indicate the potential of exosomes as therapeutic agents, the scope of drug discovery using exosomes widens. In the field of regenerative medicine, in particular, new treatments that advocate cell-free therapy using exosomes derived from MSCs, tissue stem cells, and immunocompetent cells is highly likely to be applied in clinical practice. In contrast to exosomes that exhibit benign properties, malignant exosomes are released due to a disease function through cancer cell metastasis and drug resistance. With the new understanding of cancer metastasis through exosomes, blocking exosome secretion and function of cancer cells will lead to drug discovery. Thus, exosome research has progressed at an accelerated pace, and the use of highly purified exosomes that perform their original functions is preferred. This purification technology to obtain exosomes of high purity is hypothesized to become an innovative analysis technology that will significantly change the methodology of exosome studies. 

## Figures and Tables

**Figure 1 membranes-12-01244-f001:**
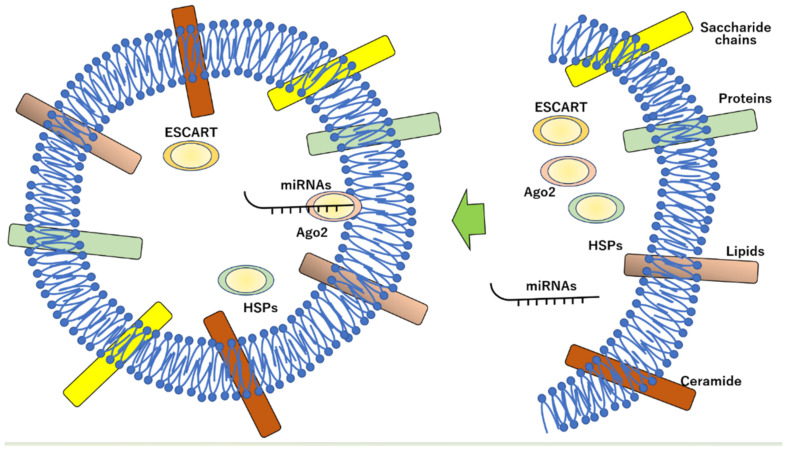
The exosome formation and structure with components. The exosome is constituted by bi-layer membranes, on which some molecules, such as cermide, lipid, prorein, and saccharide, are located. Some proteins, HSPs, Ago2, and ESCART, and miRNAs are encapsulated during formation of exosome.

**Figure 2 membranes-12-01244-f002:**
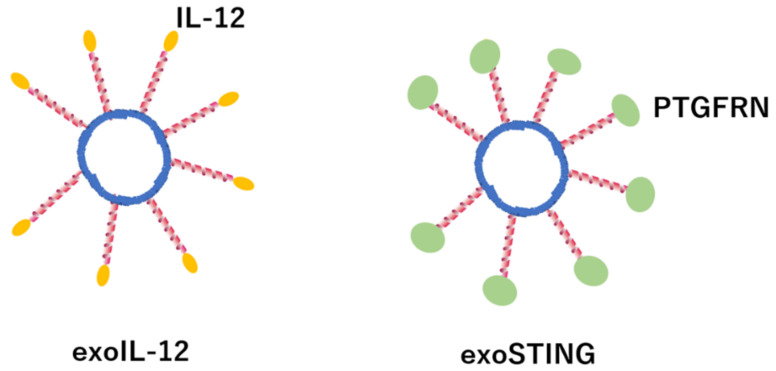
exoIL-12 is improved exosome expressed IL-12 on surface of exosome. ExoSTING in improved exosome encapsulating a STING agonist in exosomes, in which a therapeutic protein such as prostaglandin F2 receptor negative regulator (PTGFRN), is highly expressed in the exosome membrane.

**Figure 3 membranes-12-01244-f003:**
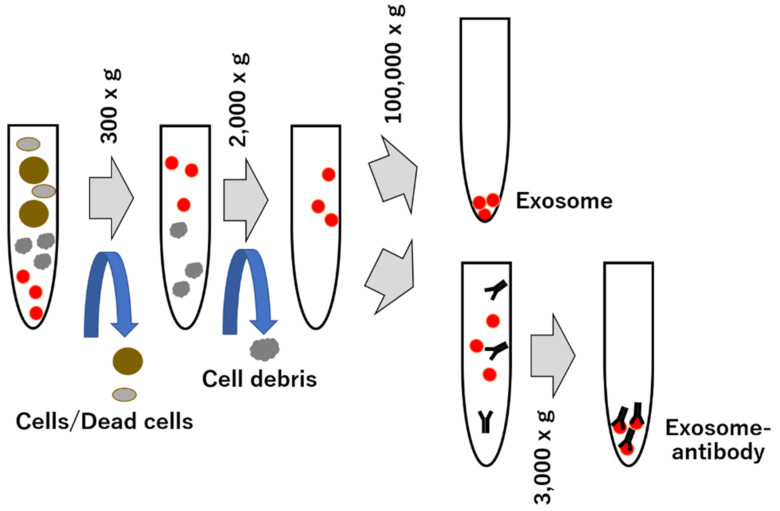
The exosome purification by ultracentrifugation and immunoprecipitation using antibody. Contents in solution from supernatant of culture cells and blood samples are isolated by sequential centrifugation with different centrifugation speeds. In addition, some antibodies for some antigens on surface membrane of exosome can bind and isolated via association between exosome and antibody.

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
