# Peer review of "Advances in Purification, Modification, and Application of Extracellular Vesicles for Novel Clinical Treatments"

_membranes, 2022, doi:10.3390/membranes12121244_

Round 1

Reviewer 1 Report

The authors summarized advances in the purification, modification, and application of extracellular vesicles (EVs) for clinical treatments. However, there are certain concerns about Figures. Therefore, I will reconsider accepting this work once the following questions are well addressed.

1. Figure 1and 2 are both vague and need additional explanation, so please provide more details.

2. There aren’t any applications of EVs for the disease treatment shown in Figures in this review. Please provide an overview of typical early articles.

Author Response

The authors summarized advances in the purification, modification, and application of extracellular vesicles (EVs) for clinical treatments. However, there are certain concerns about Figures. Therefore, I will reconsider accepting this work once the following questions are well addressed.

  1. Figure 1and 2 are both vague and need additional explanation, so please provide more details.

Thank you very much for your suggestion. According to the comments, we added some explain in legends of Figure 1 and 3.

  1. There aren’t any applications of EVs for the disease treatment shown in Figures in this review. Please provide an overview of typical early articles.

   Thank you very much for your suggestion. According to the comments, we added Figure 2.

Reviewer 2 Report

Dear Authors,

The manuscript entitled" Advances in Purification, Modification, and Application of Extracellular Vesicles for Novel Clinical Treatments" advancements in exosome purification and application as drug delivery vehicles has been described. The ms still needs improvement in some sections. My comments/suggestions to the authors are: In heading 2" constitution, characterization, and modification of exosomes": the content does not match the heading. The description covers the constitution but no modification. Changing the heading or describing the modifications of exosomes with relevant references should be considered by the authors. A distinction should be made between encapsulating drugs and genes in exosomes and modifying them. A diagrammatic depiction would be preferred. Heading 6"Application of exosomes in plant and food as DDS", there are numerous references that don't give a clear indication of what the authors are attempting to convey.

Author Response

Dear Authors,

The manuscript entitled" Advances in Purification, Modification, and Application of Extracellular Vesicles for Novel Clinical Treatments" advancements in exosome purification and application as drug delivery vehicles has been described. The ms still needs improvement in some sections. My comments/suggestions to the authors are: In heading 2" constitution, characterization, and modification of exosomes": the content does not match the heading. The description covers the constitution but no modification. Changing the heading or describing the modifications of exosomes with relevant references should be considered by the authors. A distinction should be made between encapsulating drugs and genes in exosomes and modifying them. A diagrammatic depiction would be preferred. Heading 6"Application of exosomes in plant and food as DDS", there are numerous references that don't give a clear indication of what the authors are attempting to convey.

Thank you very much for your suggestion. According to the comments, we corrected heading 2 to 2. Constitution and characterization of exosomes. Also, according to the comments, we corrected references as follows, 265, 359-362].

Round 2

Reviewer 1 Report

The manuscript can be accepted with this current format. Thanks.

Reviewer 2 Report

The manuscript can now be accepted.